# Understanding IGF-II Action through Insights into Receptor Binding and Activation

**DOI:** 10.3390/cells9102276

**Published:** 2020-10-12

**Authors:** Andrew J. Blyth, Nicholas S. Kirk, Briony E. Forbes

**Affiliations:** 1Department of Medical Biochemistry, Flinders Health and Medical Research Institute, Flinders University, Bedford Park 5042, Australia; andrew.blyth@flinders.edu.au; 2Walter and Eliza Hall Institute of Medical Research, Parkville 3052, Victoria, Australia; kirk.n@wehi.edu.au; 3Department of Medical Biology, Faculty of Medicine, Dentistry and Health Sciences, University of Melbourne, Parkville 3050, Victoria, Australia

**Keywords:** IGF-II, insulin-like growth factor, IGF-1R, insulin receptor, IR-A, structural studies, receptor activation

## Abstract

The insulin-like growth factor (IGF) system regulates metabolic and mitogenic signaling through an intricate network of related receptors and hormones. IGF-II is one of several hormones within this system that primarily regulates mitogenic functions and is especially important during fetal growth and development. IGF-II is also found to be overexpressed in several cancer types, promoting growth and survival. It is also unique in the IGF system as it acts through both IGF-1R and insulin receptor isoform A (IR-A). Despite this, IGF-II is the least investigated ligand of the IGF system. This review will explore recent developments in IGF-II research including a structure of IGF-II bound to IGF-1R determined using cryo-electron microscopy (cryoEM). Comparisons are made with the structures of insulin and IGF-I bound to their cognate receptors. Finally discussed are outstanding questions in the mechanism of action of IGF-II with the goal of developing antagonists of IGF action in cancer.

## 1. Introduction

The insulin-like growth factor (IGF) system controls metabolic and mitogenic responses in mammalian cells and importantly regulates embryonic growth and development as well as adult growth [1]. The IGF system is regulated by three structurally similar ligands, IGF-I, IGF-II and insulin (Figure 1). These ligands act via one or more of the three related receptor tyrosine kinases: the two splice variants of the insulin receptor (IR-A and IR-B) and the type 1 Insulin-like growth factor receptor (IGF-1R). IR-B signaling is responsible for the classic IR metabolic activities. IGF-II is unique in that it can activate both IGF-1R and IR-A to promote cell growth and survival. However, of the three ligands, the molecular mechanisms underlying IGF-II action are the least understood. For this reason, this review will focus on IGF-II. There is some evidence to suggest that IGF-II, IGF-I and insulin can promote shared and unique signaling outcomes through IGF-1R and IR [2,3]. However, IGF-II specific actions are generally attributed to tissue specific expression. This review will highlight new discoveries regarding IGF-II, including a cryo-electron microscopy (cryoEM) structure of IGF-II bound to IGF-1R that has provided vital information on the structure and function of IGF-II.

IGF-II plays important roles in fetal growth and development, when it is most abundant [4,5]. Notably, IGF-II fetal plasma concentrations are several fold higher than that of IGF-I [6]. Knockout of *Igf2* leads to a 60% reduction in weight at birth [7]. IGF-II serum concentrations in many mammalian species decline rapidly after birth [8,9,10]. Interestingly, in adult mice, IGF-II serum levels are barely detectable, whereas in humans it is the more abundant IGF ligand [11]. In humans, the *IGF2* is maternally imprinted and only expressed from the paternal allele. Gain of methylation at the regulatory H19 locus on the paternal allele causes underexpression of IGF-II and results in undergrowth syndromes (Russell-Silver syndrome), which can include a variety of phenotypes including prenatal growth deficiency, facial dysmorphic features and developmental delay [12,13]. Alternatively, overexpression of IGF-II can produce an overgrowth syndrome (Beckwith–Wiedemann syndrome), which can include macroglossia, macrosomia, and abdominal wall defects [12,13]. For a detailed review of the genetic regulation of IGF-II expression in physiology and disease refer to [14].

At the tissue level, IGF-II promotes cell growth and survival. It regulates bone growth by promoting proper timing of chondrocyte maturation and perichondrial cell differentiation and survival [15]. Overexpression of *Igf2* in smooth muscle and pancreatic beta cells results in the development of cardiovascular defects and type 2 diabetes [16,17]. Conversely, knockout of placental *Igf2* leads to reduced placental growth and fetal growth restriction [18]. IGF-II is most abundant in the fetal and adult brain, primarily produced by the choroid plexus but also the leptomeninges and endothelial cells [19,20,21,22,23]. IGF-II has been identified in cerebral spinal fluid and has been found to promote neurogenesis in the subventricular and subgranular zone of the adult brain [24,25,26]. Several investigations have also identified that IGF-II promotes stem cell self-renewal through activation of IR-A. For example, IGF-II:IR-A signaling supports neural stem cell maintenance and the expansion of neural progenitor cells [27]. This role in stem cell renewal extends to other tissues, as identified using stem cell specific knockout of *Igf2* in young adults in which growth of intestinal stem cells is also inhibited [28].

IGF-II action is highly regulated by its interaction with soluble IGF binding proteins, including IGF-II specific IGFBP-6. IGFBPs retain IGF-II in circulation and deliver it to target tissues [29]. In addition, the type 2 IGF receptor (IGF-2R, also called cation-independent mannose-6-phosphate receptor) is responsible for the control of circulating IGF-II levels, by binding to IGF-II with high affinity and targeting it for lysosomal degradation [30,31].

## 2. IGF-II and Cancer

It is well established that abnormal function of the IGF system promotes growth and metastasis of the 3 most commonly diagnosed cancers: breast, prostate and colorectal [32,33,34]. It also promotes growth and survival of brain, thyroid and ovarian cancers among others [11,14]. Specifically, dysregulation of IGF-II expression has been associated with cancer progression [11]. IGF-II expression is often upregulated in these cancers [33,34,35] and often results in both autocrine and paracrine effects [36]. For example, in the MDA-MB-157 breast cancer cell line, autocrine production of IGF-II stimulates cell growth though IR-A activation while expression in stromal and epithelial tissue of breast cancer specimens acts in both autocrine and paracrine manners [37]. Loss of imprinted IGF-II expression has been documented in many forms of cancer, leading to increased levels of intratumoural IGF-II, thereby promoting cell growth and tumorigenesis [34,38,39]. Interestingly, the mechanism by which loss of imprinting occurs has recently been investigated and found to involve overexpression of an intronic miRNA (miR-483-5p) found within the *IGF2* gene [40]. miR-483-5p increases IGF-II transcription at the fetal promoter [40].

In cancer, IGF-II can act via IGF-1R and/or IR-A and these autocrine/paracrine signaling loops are regularly observed [41]. IGF-1R, which promotes cell growth and survival, is also commonly upregulated in cancers such as breast, colorectal and prostate cancer [35,41,42]. In contrast to IR-B that signals through metabolic pathways, IR-A has mitogenic signaling capabilities that are important during development when IR-A is most abundantly expressed [33]. IR-A is only expressed at very low levels in most adult cells [43]. However, in malignant cells, including breast, thyroid, colon and prostate cancer, IR is over expressed, and IR-A is the predominant isoform [33,44]. IGF-II:IR-A signaling also supports maintenance of tumour stem and progenitor cells [45,46]. Concomitant upregulation of both IGF-II and IR-A signaling thus provides cancer cells and tumour stem cells with an additional growth and survival mechanism [11]. 

## 3. IGF-II Signaling 

The biological processes that IGF-II promotes result from activation of signaling pathways through its binding to the extracellular region of IR-A or IGF-1R. The overall mechanisms of binding of IGF-II, IGF-I and insulin to IGF-1R and IR are conserved. Receptor binding results in structural rearrangement of the receptor (further discussed below) causing autophosphorylation of the tyrosine kinase (TK) domains on the intracellular region of the receptor [47,48]. Extensive studies conducted by Cabail et al. [49] have determined that in the unbound state, each monomer is autoinhibited by self-interaction of the activation loop within its TK active site, thereby precluding the binding of ATP. Upon ligand binding, structural rearrangement occurs allowing the juxtamembrane (JM) domain of one monomer to interact with the TK domain of the opposite monomer. This releases the autoinhibitory state and allows for the binding of ATP and subsequent substrate phosphorylation.

The first signaling step upon IGF-II, IGF-I, and insulin binding to their cognate receptors involves phosphorylation of three tyrosine residues within the activation loop of the TK domain (IGF-1R: Y1131, Y1135, and Y1136 and IR: Y1158, Y1162 and Y1163) [47,50]. Subsequently, residue Y950 (IGF-1R) or its equivalent Y960 (IR) is phosphorylated [51]. This creates a docking site for IR substrates (IRS) and Shc (Figure 2), which are then phosphorylated [52]. Subsequent to receptor activation, two main signaling pathways are activated, the phosphoinositide 3-kinase (PI3K)-Akt/protein kinase B (PKB) pathway, responsible for metabolic responses and the Ras-mitogen-activated protein kinase (MAPK) pathway, resulting in mitogenic responses (cell growth, differentiation, and gene expression) [48,53,54].

## 4. How does IGF-II Bind and Activate IGF-1R and IR-A?

In order to understand how IGF-II promotes normal cell growth and survival and to develop ways to inhibit its action in cancer, a detailed knowledge of the molecular mechanisms underlying IGF-II receptor binding and activation is required. Our understanding so far has largely been derived through site-directed mutagenesis and comparative structural studies, with a recent cryoEM study revealing the structure of IGF-II bound to IGF-1R. The details of our current understanding will now follow.

### 4.1. IGF-II Structure

IGF-II is a 67 amino acid single chain polypeptide with sequence and structural similarity to IGF-I (70 amino acids) and insulin (51 amino acid two-chain peptide) (Figure 1). Sequence alignments of the IGFs and insulin (Figure 1a) reveal 50% sequence homology between the B- and A-domains of the IGFs and the equivalent domains of insulin [1]. Three intrachain disulfide bonds hold together the specific three-dimensional structure, which comprises three α-helices (Figure 1b). IGF-I and IGF-II each comprise four domains: B, C, A and D [55]. Insulin, in contrast, is a two-chained mature protein composed of A and B domains joined together by two inter-chain disulfide bonds and having one intra-chain disulfide bond within the A chain (Figure 1b) [56].

IGF-II contacts the receptor through two surfaces originally defined by site-directed mutagenesis that are named site 1 and site 2. Equivalent residues of IGF-I and insulin are involved in binding IGF-1R and IR, respectively (Table 1).

### 4.2. Receptor Structure, Mechanism of Binding and Activation

IR-A, IR-B and IGF-1R are similar in amino acid sequence and structure (Figure 3a). The two IR isoforms differ by the expression of exon 11, which consists of 12 amino acids that are absent in IR-A splice variant. The receptors are disulfide-linked (αβ)2 homodimers and the extracellular domains of each αβ monomer assemble in an anti-parallel, Λ-shaped conformation, generating two equivalent ligand binding regions. In the apo (unbound) state, the sites of membrane entry are situated far apart, thereby holding the intracellular tyrosine kinase in an inactive monomeric state (Figure 3b left) [57,58]. Site-directed mutagenesis and structural studies have identified two binding surfaces within each binding region (site 1 and site 2) that represent high- and low-affinity binding sites, respectively. Upon ligand binding, the receptors undergo extensive structural change, whereby the FnIII stalks come close together, permitting dimerization of the intracellular region to release the autoinhibition of the TK domains (Figure 3b right). Notably, such a conformation is as predicted by Kavran et al. [59] to be essential for IGF-1R activation.

Molecular detail of receptor binding in the extracellular domain has been derived from a series of crystallographic and cryoEM studies of IGF-1R and IR in the holo and soluble ectodomain forms. The α-chain *C*-terminal (αCT) helices of each monomer lie on the L1 surface of the opposing monomers to form site 1 [57,58]. The αCT shifts to accommodate the ligand, which makes contact via its site 1 residues [60,61]. As defined by Weis et al. [62], site 1 contacts made between the ligand and the receptor L1 and αCT domains [62,63,64]. In the case of IGF-II and IGF-I binding IGF-1R as well as insulin binding IR, the residues identified in site-directed mutagenesis studies correspond to those involved in this site 1 interaction (Table 1). Several additional residues were revealed in these structures to contact the L1 and αCT and can now be defined as site 1 residues (Table 1). 

Recently, a structure of the IGF-II:IGF-1R complex was determined using cryoEM to an average maximum resolution of 3.2 Å (Figure 4b) [63]. The site 1 ligand binding interaction is similar to the previous insulin:IR and IGF-I:IGF-1R structures [64,69,70]. The IGF-II molecule contacts the L1, L2, αCT’, and FnIII-1’ domains within the head region of the receptor (Figure 4c) [63]. The L1-CR + (αCT’) module folds to the top of the receptor, permitting sparse interactions between IGF-II and the membrane-distal loops of FnIII-1’, facilitated by an outward rotation of domain L2 from its location in the apo ectodomain. The αCT’ helix on the L1 domain surface threads through the IGF-II C-domain loop (residues 33–40). The *C*-terminal segment of the IGF-II B-domain is displaced from the core of the ligand (in the unbound state) and engages with the receptor to make the site 1 interaction. The B chain of IGF-II is stabilized by an interaction between IGF-II residue Arg30 and the hydroxyl group of IGF-1R residue Tyr28 and possibly a salt bridge between IGF-II residue Arg38 and IGF-1R residue Glu305. The ligand forms a ‘clip’ on the extended αCT helix in the active conformation, stabilizing a tight interaction between L1-CR-L2 and αCT’ with only sparse interactions between the ligand and FnIII-1’ (Figure 4c) [63].

This overall J-shaped conformation that brings the FnIII stalks together was seen previously in the Weis et al. insulin:IR and Li et al. IGF-I:IGF-1R studies [62,64]. The IGF-II:IGF-1R complex structure [63] was determined using a similar leucine-zippered receptor (IGF-1RZip) to that of the insulin receptor used in the Weis et al. study [62]. The general topology of IGF-1RZip:IGF-II and IRZip:insulin structures also reflects that of the recently reported holoIGF-1R:IGF-I structure [64], providing further evidence that this is the common activated conformation. The asymmetry observed in the activated structure is necessary for negative co-operativity, a hallmark of both IGF-1R and IR ligand binding summarized in a ‘harmonic oscillator model’ by Kiselyov et al. [71], whereby binding of a second ligand (to the unoccupied receptor binding pocket) accelerates the dissociation of the first bound ligand.

Looking at the molecular detail of the IGF-II:IGF-1R complex structure confirms that the IGF-II site 1 residues identified by site-directed mutagenesis interact with IGF-1R site 1 (summarized in Table 1 and Figure 4). This involves side chains of residues of the B-domain (Cys9, Leu13, Val14, Asp15, Leu17, Gln18, Asp23, Phe26, Tyr27, Phe28, Ser29, and Arg30) contacting the L1 and αCT’ segment and side chains of A-domain residues (Ile42, Val43, Glu44, Phe48, Thr58, Tyr59, and Thr62) contacting the αCT’ segment (but not the L1). These side chain interactions are similar in the IRZip:insulin and holoIGF-1R:IGF-I structures as the ligand sequences are highly conserved in these regions (Table 1).

The major difference in the structures of IGF-1R ectodomain-bound IGF-II and IGF-I occurs in the respective growth factor C-domains. In the receptor complex, the IGF-II C-domain residues 33–36 are disordered, as are the adjacent receptor CR domain residues 258–265, suggesting that the C-domain is too short to form stable interactions with the receptor in this region (Figure 5a) [63]. By contrast, the C-domain of IGF-I in holoIGF-1R:IGF-I is relatively well ordered, with IGF-I residue Tyr31 in its distal loop engaging receptor residues Pro5 and Pro256 (Figure 5a) [64]. Although the resolution of the structure is low at IGF-I residues Arg36 and Arg37, they appear to contact IGF-1R L2 domain. With no equivalent to Tyr31, the IGF-II C-domain instead appears to be stabilized by self-interactions (a salt bridge with IGF-II residue Glu45 near the *N* terminus of the first helix of the IGF-II A-domain, and a polar interaction with the IGF-II residue Ser39).

An as yet unexplained observation is the limited correlation of the site-directed mutagenesis data for IGF-II site 2 residues (Table 1) and their involvement in binding in the IGF-II:IGF-1R complex structure (Figure 5b). Of the residues defining site 2 by site-directed mutagenesis, only Glu12 appears to contact the receptor FnIII-1’ in this activated conformation (Figure 5b). In addition, IGF-II B-domain residues Glu6, Thr7, Cys9 and A-domain residues Cys47 and Phe48 are seen to contact the FnIII-1’ in the IGF-II:IGF-1R complex structure, thereby completing the definition of site 2 (Table 1). A similar conundrum was revealed by IGF-I:IGF-1R and insulin:IR complex structures and their corresponding site-directed mutagenesis data (Table 1).

For both IGF-I:IGF-1R and insulin:IR complexes additional structures have been described that have led to a proposed transient interaction of the ligand with a different site on the receptor. This may represent the first site of contact for the ligand or an intermediate site to facilitate conformational change of the ligand and receptor (schematically represented in Figure 3b, middle panel). In the case of insulin, cryoEM structures of insulin-saturated IR constructs [69,70] identified potential transient binding sites on the FnIII-1’ spanning residues Tyr477-488 and 552-554 involving all insulin site 2 residues (Table 1). Such a site has not been reported for IGF:IGF-1R complexes. For IGF-I:IGF-1R, the first ligand-bound ectodomain structure was determined by X-ray crystallography by Xu et al. [58]. This structure was determined by ligand soaking in apo crystals, resulting in an induced fit of IGF-I to the L1- αCT’ binding site. In this structure, the receptor remained in the “apo/legs apart” conformation without the major J-shaped rearrangement. Additional FnIII-2’ contacts (residues 788-792) were observed that involved essentially all IGF-I site 2 residues identified by site-directed mutagenesis except Glu9 and Asp12. It is possible that this interaction represents an IGF-1R transient binding site and suggests a major difference in the activation mechanism between the two receptors. Whether these transient interactions also occur for IGF-II on IGF-1R and IGF-II on IR-A remains to be determined.

In summary, whilst IGF-II binds and activates IGF-1R through a similar mechanism to IGF-I, there are some significant differences that likely explain their different binding affinities. Notably the C-domain interactions are quite different, with IGF-II barely making receptor contact, whereas IGF-I C-domain contributes to binding affinity through several contacts. How this influences ligand specific signaling outcomes is still not understood. Importantly, no structure of IGF-II bound to IR-A has been reported.

## 5. Conclusions and Implications of Structural Information for Developing Treatments for Disease

IGF-II plays a fundamental role in mammalian growth and fetal development. It is an important regulator of bone growth and promotes cellular growth and survival. While IGF-II is the least investigated ligand of the IGF system, the recently determined structure of IGF-II bound to IGF-1R has certainly advanced our understanding of the mechanism of IGF-II binding and activation. This structural information has confirmed that upon IGF-1R engagement, the receptor undergoes major structural rearrangement, from an open Λ-shape conformation to a J-shaped structure where the legs of the receptor are brought into contact in the active signaling conformation of the receptor. Comparison of IGF-I and IGF-II bound to IGF-1R confirmed that the C-domain of IGF-I contacts the receptor, whereas IGF-II lacks an equivalent contact. While site 1 contacts of IGF-II are in accordance with mutagenesis data, only one site 2 residue is seen to contact the receptor (Glu12) as observed in IGF-I (Glu9) and insulin bound to IR (HisB10). The remaining residues identified by mutagenesis as contacting the receptor may be involved in transient interactions with the receptors. The same transient interaction is expected with IGF-II binding; however, this is yet to be observed. 

A detailed understanding of how IGF-II engages with its receptors and confers downstream signaling activation is essential in developing drug therapies that target IGF action in cancer. The relatively minor role of IGF-II in adult cell function means that blocking this pathway as a cancer therapy may have little effect on healthy adult cells whilst slowing cancer cell growth. Currently, most approaches target IGF action by directly blocking binding to IGF-1R:IGF-1R antibodies inhibit ligand binding and stimulate receptor internalisation [34]. Such inhibitors have been shown to reduce growth of IGF-II dependent cancers. However, increases in IGF-II:IR-A signaling can give rise to resistance to treatment [72,73], highlighting the need for inhibitors of IGF-II acting via both IGF-1R and IR-A and the need for structural data of IGF-II bound to IR-A. Such studies will further inform on how IGF-II is uniquely capable of binding and activating both IR-A and IGF-1R with high affinity and will suggest strategies to design inhibitors or allosteric regulators for the treatment of IGF-1R/IR-A regulated disease.

## Figures and Tables

**Figure 1 cells-09-02276-f001:**
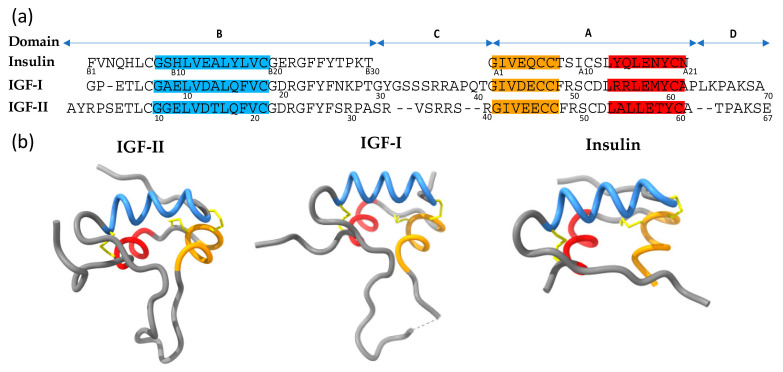
(**a**) Sequence alignment of IGF-II, IGF-I and insulin proteins. Domains are indicated above. Each peptide has three alpha helices; B-chain helix 1 (Blue), A-chain helix 2 (orange) and A-chain helix 3 (red). Residue numbers are indicated below each sequence. (**b**) Ribbon structures of the IGF and insulin proteins (PDB: 1IGL, 1GZR and 1MSO respectively). The three disulfide bonds in each protein are represented in yellow.

**Figure 2 cells-09-02276-f002:**
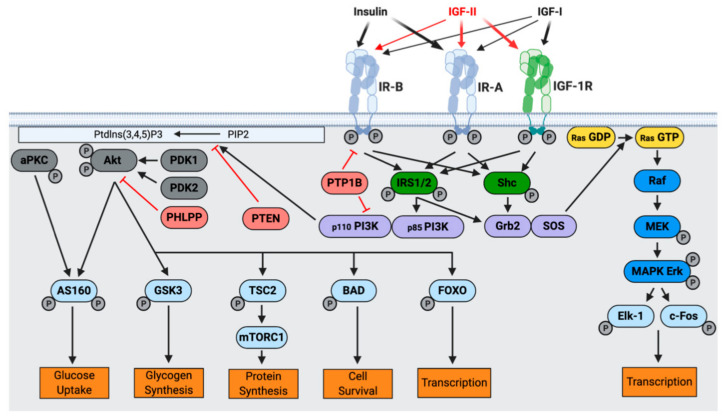
The Insulin and IGF system. Insulin, IGF-I and IGF-II bind with different affinities to IR-B, IR-A and IGF-1R (indicatd by thickness of arrows). IGF-II binds with high affinity to both IGF-1R and IR-A, and with low affinity to IR-B. Upon receptor binding, a structural change leads to activation of the intracellular tyrosine kinase domain and autophosphorylation (indicated by P). IRS1/2 and Shc adapter proteins are recruited and two main signaling pathways are activated: the Akt/PKB and the Ras/MAPK pathways. Metabolic and mitogenic activities are promoted, respectively. (Adapted from: [48]).

**Figure 3 cells-09-02276-f003:**
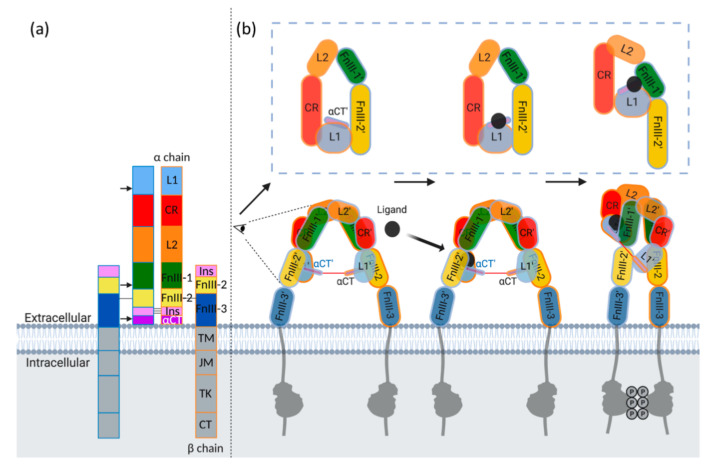
(**a**) Domain structure of IGF-1R and IR receptor tyrosine kinases. Individual αβ monomers are indicated by blue and orange outline. IGF-1R and IR have a high degree of sequence homology and therefore comprise the same domains: first and second leucine-rich repeat domains (L1 and L2), cysteine-rich domain (CR), first, second and third type-III fibronectin -like domains (FnIII-1, 2, and 3), insert domain (ID), α-chain *C*-terminal domain (αCT), transmembrane domain (TM), juxtamembrane domain, (JM), tyrosine kinase (TK), *C*-terminal domain (CT). Arrows indicate regions involved in ligand binding. (**b**) Schematic representation of the mechanism of ligand binding. Side view of binding pocket shown in blue dotted box. In the unbound (apo) state (left) the receptor forms an open Λ-shape with FnIII-3 legs situated far apart. Ligand binding is likely to involve a transient interaction (middle) followed by major structural rearrangement forming a J-shape active conformation (right) where the FnIII-3 legs of the receptor are in close proximity. In turn, a structural change occurs in the intracellular domains leading to autophosphorylation by the TK.

**Figure 4 cells-09-02276-f004:**
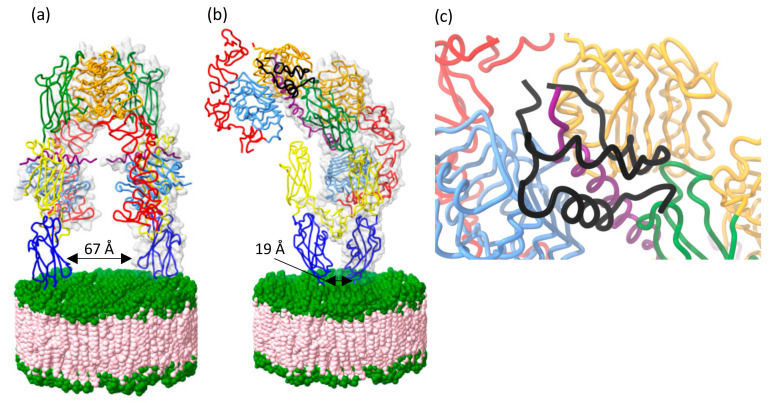
(**a**) Crystal structure of the unbound (apo) IGF-1R ectodomain (PDB: 5U8R). In the apo state the FnIII legs of the receptor are positioned far apart forming an open Λ-shape. Upon IGF-II (black) binding, a major structural rearrangement occurs resulting in a J-shape conformation of the receptor where the FnIII legs are in close proximity. (**b**) The activated conformation (PDB: 6VWI and 6VWJ) is stabilized by the ligand clipping the αCT and L1 domains together, interactions through site 2 on FnIII-1’, and potential salt bridges in the head region facilitated by ligand binding (between Glu687’ (αCT’) and Arg335 (domain L2), between residues Glu693’ (αCT’) and Arg488’ (domain FnIII-1’), and between residues Lys690’ (αCT’) and Asp489’ (domain FnIII-1’)). (**c**) Zoom-in of the site 1 ligand binding region between IGF-II and IGF-1R involving IGF-II B-domain residues; Cys9, Leu13, Val14, Asp15, Leu17, Gln18, Asp23, Phe26, Tyr27, Phe28, Ser29, and Arg30 and the side chains of receptor domain L1 residues Pro5, Ile7, Asp8, Arg10, Asn11, Leu33, Ser35, Ly36, Phe58, and Arg59, and the side chains of receptor αCT’ residues His697’, F701’, Val 702’, and Pro705’. The IGF-II A-domain contacts the receptor αCT’ domain (and not domain L1), with the interaction mediated by the side chains of IGF-II residues Ile42, Val43, Glu44, Phe48, Thr58, Tyr59, and Thr62 and the side chains of receptor αCT’ residues Lys690’, Glu694’, His697’, Asn698’, Phe701’, Val702’, Pro703’, and Arg704’. (**a**–**c**) coloured as in Figure 3.

**Figure 5 cells-09-02276-f005:**
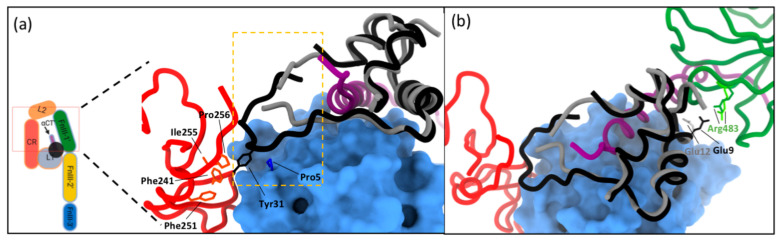
(**a**) Interaction of the C-domain (shown in yellow box) of IGF-II (grey) and IGF-I (black) with IGF-1R (PDB: 6VWI and 6PYH, respectively). Domains of IGF-1R coloured as in Figure 3. Residues of the CR and L1 domain engage with residue Tyr31 of IGF-I. Contact is also made between C-domain residues Arg36 and Arg37 of IGF-I and the L2 domain. There is no equivalent C-domain residue in IGF-II. (**b**) Site 2 contacts involve residues Glu12 of IGF-II and Glu9 IGF-I which contact FnIII-1 domain residue Arg483 (dark green represents the IGF-1:IGF-1R structure and light green the IGF-II:IGF-1R structure). (**a**,**b**) coloured as in Figure 3.

**Table 1 cells-09-02276-t001:** Binding site 1 and 2 residues of insulin, IGF-I and IGF-II. Residues identified as contacting the receptor through mutagenesis studies (coloured blue), structural studies (red) or both (black). * Residues observed to make transient contact with the FnIII-1 domain of IR [64]. ^#^ Residues observed to make transient contact with the FnIII-1 domain of IGF-1R [57]. ^a^ Asp45Ala IGF-I mutant results in 3-fold decrease in binding affinity [65].

	IGF-II		IGF-I [57,65]		Insulin [64]	
**Site 1**	Cys9	[63]	Cys6	[64]	CysB7	[62]
	Leu13		Leu10		LeuB11	
	Leu17		Leu14		LeuB15	
	Asp23		Asp20		GluB21	
	Ser29		Asn26		ProB28	
	Arg30		Lys27			
	Thr58		Met59		AsnA18	
	Thr62		Lys65			
	Val14	[1,63]	Val11	[1,64]	ValB12	[62,65]
	Gln18		Gln15		TyrB16	
	Gly25		Gly22		GlyB23	
	Phe26		Phe23		PheB24	
	Tyr27		Tyr24		PheB25	
	Phe28		Phe25		TyrB26	
			Tyr31			
			Arg36			
			Arg37			
	Gly41		Gly42		GlyA1	
	Ile42		Ile43		IleA2	
	Val43		Val44		ValA3	
	-		Asp45 ^a^		GluA4	
	Glu45		Glu46		GluA5	
	Phe48		Phe49		ThrA8	
	Tyr59		Tyr60		TyrA19	
	Ala61		Ala62		AsnA21	
**Site 2**	Glu6	[63]	Glu3	[1]	GlnB4	[62]
	Thr7	[63]	Thr4	[64]	HisB5	[62]
	Cys9	[63]	Cys6	[64]	CysB7	[62]
	Glu12	[63,66]	Glu9 ^#^	[64,67]	HisB10 *	[62,68]
	Asp15	[66]	Asp12	[64,67]	GluB13 *	[62,68]
	Phe19	[66]	Phe16 ^#^	[64,67]	LeuB17 *	[62,68]
	Cys47	[63]	Cys48	[64]	CysA7	[62]
	-		-		SerA12	[62,68]
	Leu53	[66]	Leu54 ^#^	[67]	LeuA13 *	[62,68]
	Glu57	[66]	Glu58 ^#^	[67]	GluA17 *	[62,68]

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
