# Peer review of "Understanding IGF-II Action through Insights into Receptor Binding and Activation"

_cells, 2020, doi:10.3390/cells9102276_

Round 1

Reviewer 1 Report

Major comments

  1. After IGF-II bind to IGF-1R, is its downstream signals different from IGF-I ?
  2. After IGF-II bind to IGF-1R, is its effects different from IGF-I ?
  3. Why the authors want to inhibit IGF-II only, but not both IGF-I and IGF-II ?

Minor comments

  1. What is (REF) on line 44 ?
  2. The authors should not use abbreviation on Abstract; cryo-EM on line 21.

Reviewer 2 Report

This review written by one of the leading groups in the biology and structural biology of the IGF system is timely and informative, and clearly written. IGF-II has been the most poorly studied member of the insulin-like peptide family primarily because its expression becomes mostly extinct after birth in rodents, while in adult humans its circulating level is higher than IGF-I. Hopefully this review will boost interest in the physiology of this peptide in adult life.

I am a bit surprised that in a review on “insights into receptor binding and activation” no attention is given to the well documented mechanism of activation of the receptor tyrosine kinase domain through dimerization and phosphorylation of the kinase inhibition loop. The extensive structural work of Stevan Hubbard and colleagues on insulin and IGF-I RTKs is not even cited once (see for example Cabail MZ et al. Nat Commun 6, 6406 (2015) and references therein).

The work of Kavran JM et al. on IGF-IR activation would also deserve to be cited (eLife 10.7754/eLife. 03772).

Specific comments

Page 2 lines 44-45: “IGF-II serum concentrations decline rapidly after birth (6,7)”. It should be specified that this is in mice. References 6 and 7 are incorrect. Reference 6 compares IGF-I levels measured by KIRA bioassay with five RIAs. Reference 7 deals with IGFs in human skin interstitial fluid.

Page 2 line 61: “… leptomeninges and endothelial cells (7,15)”: reference 7 is incorrect, it deals with skin interstitial fluid.

Page 3 line 70: “the type 2 IGF receptor (IGF-2R) …”: add: “(also called cation-independent mannose-6-phosphate receptor)”.

Page 3 line 84: “… an intronic miRNA (miR-483-5p) found within the IGF2 gene (32)”: reference 32 is incorrect, it deals with the transcription factor E2f3 and not with the miRNA.

Page 6, Figure 3b: The depiction of the intracellular part of the receptor and of RTK activation is not satisfying. The proportion of the RTK domain compared to that of the extracellular domain is wrong, see for example Fig. 1 of Hubbard SR and Miller WT, eLife 2014;3:e04909. The juxtamembrane domains above and below the membrane are missing. The kinase activation mechanism deserves better depiction that an “explosion” labelled “activation”.

Page 7 Table 1. ThrA8 does not appear to be in site 1 of the insulin molecule as previously thought, see Fig. 4 and discussion in ref. 58. Insulin residues in site 2: the structure in ref. 57 supports only His B10 and Glu B13. See also (and cite) ref. 58, table 1.

References: The journal titles are sometimes abbreviated, sometimes spelled out. Words in titles are sometimes capitalized, sometimes not. This should be harmonized.
